# Economic Perspective of Cancer Care and Its Consequences for Vulnerable Groups

**DOI:** 10.3390/cancers14133158

**Published:** 2022-06-28

**Authors:** Joerg Haier, Juergen Schaefers

**Affiliations:** Comprehensive Cancer Center Hannover, Hannover Medical School, Carl-Neuberg-Str. 1, 48149 Hannover, Germany; schaefers.juergen@mh-hannover.de

**Keywords:** economy, financial burden, universal health coverage, vulnerable groups, cancer

## Abstract

**Simple Summary:**

For cancer patients, many different reasons can cause financial burdens and economic threads. Sociodemographic factors, rural/remote location and income are known determinants for these vulnerable groups. This economic vulnerability is related to the reduced utilization of cancer care and the impact on outcome. Financial burden has been reported in many countries throughout the world and needs to be addressed as part of the sufficient quality of cancer care.

**Abstract:**

Within healthcare systems in all countries, vulnerable groups of patients can be identified and are characterized by the reduced utilization of available healthcare. Many different reasons can be attributed to this observation, summarized as implementation barriers involving acceptance, accessibility, affordability, acceptability and quality of care. For many patients, cancer care is specifically associated with the occurrence of vulnerability due to the complex disease, very different target groups and delivery situations (from prevention to palliative care) as well as cost-intensive care. Sociodemographic factors, such as educational level, rural/remote location and income, are known determinants for these vulnerable groups. However, different forms of financial burdens likely influence this vulnerability in cancer care delivery in a distinct manner. In a narrative review, these socioeconomic challenges are summarized regarding their occurrence and consequences to current cancer care. Overall, besides direct costs such as for treatment, many facets of indirect costs including survivorship costs for the cancer patients and their social environment need to be considered regarding the impact on vulnerability, treatment compliance and abundance. In addition, individual cancer-related financial burden might also affect the society due to the loss of productivity and workforce availability. Healthcare providers are requested to address this vulnerability during the treatment of cancer patients.

## 1. Introduction

Currently, worldwide, 17.5 million cancer cases and 8.7 million deaths occur. During the last decade, their incidence raised by 33%, with population aging contributing 16%, population growth contributing 13% and changes in age-specific rates contributing 4% [1]. This transitional process in cancer care requirements [2] will become even more important during the next two decades, especially in countries facing intensive epidemiological changes [3]. These effects are not only related to the acquisition of environmental and lifestyle risk factors but are also determined by regional differences in vulnerability, such as those caused by genetic background, race, microbiomes and cultural aspects, among others. In this context, the socioeconomic framework and financial burden of cancer care will likely gain much more importance, especially in vulnerable groups and for infrastructural development [4,5]. This economic burden of cancer contains expenditures on care in the primary, outpatient, emergency and inpatient settings, along with drugs. In addition, due to the improvements of cancer care, increasing cancer survivorship needs to be included. Furthermore, indirect costs due to lost earnings after premature death or unemployment/cancer-related disabilities, family caregiving and costs associated with individuals who temporarily or permanently left employment because of illness contribute to the overall economic burden of cancer [6]. Worldwide, these financial consequences vary in a wide range between the countries. For example, across EU countries, the annual cancer-related costs accounted for EUR 102 per citizen, on average, but varied to a large extent between EUR 16 per person in Bulgaria to EUR 184 per person in Luxembourg [2]. The impact of various cancer entities on these costs also varies considerably, which has been shown for malignant blood disorders [7] and bladder cancer [8]. Interestingly, widely varying healthcare costs were found for countries with similar gross domestic product per capita (GDP). [9] These differences are related to a major extent but are not limited to the willingness to pay by either the society or the individual cancer patient—depending on the healthcare system.

In a recent review, Yong et al. [10] concluded that the individual willingness is intensively related to the expected outcome of cancer care, such as quality-adjusted life year (QALY), 1-year survival, quality of life (QoL) improvement and pain reduction. Economic development indicators contribute 4% of variation and increase the overall variation in incidence and mortality in breast cancer by approximately 5% [11]. However, similar empirical analyses investigating the priority of cancer care within the economic framework of national healthcare systems are not yet available. Since the economic burden for individuals is a key determinant of vulnerable groups in healthcare, we summarize the currently available literature on its impact on cancer care. Although vulnerable groups in this regard can be found to be more likely in low and middle income countries (LMIC), they are also present in industrial populations. Universal healthcare coverage criteria (UHC: availability, acceptability, accessibility, affordability, quality of care) were used as references.

Cancer patients can experience financial difficulties even within a publicly funded healthcare system [12]. These problems faced by cancer patients in Western countries have been widely explored, mostly for the Americas and the Western Pacific WHO regions [13], but they may not be applicable in other countries due to sociocultural differences [14,15]. Especially, evaluations from LMICs have rarely been found yet. In this narrative review, we critically summarize the available analyses regarding the economic perspective of cancer care for vulnerable groups. The current constraints due to the COVID-19 pandemic are out of the scope of this review.

## 2. Materials and Methods

A literature search was conducted using the items “cancer”, “economic burden”, “economy”, “vulnerable” and “LMIC” (220 results). These publications were screened regarding the relationship to the UHC criteria, and papers without relation to cancer care and economic burden were eliminated (*N* = 139). The remaining papers (*N* = 81) were divided into a group containing quantitative or qualitative research data (*N* = 33) and publications focusing on non-empirical approaches (*N* = 48). In addition, secondary literature (*N* = 28) was used to supplement specific aspects of financial consequences (Figure 1).

Data on financial difficulties or catastrophic health expenditures were extracted if available in the publications. The World Bank separates countries into four income categories by income using GDP per capita [16], and this was applied: Low Income (LIC), Lower-Middle Income (LMIC), Upper-Middle Income (UMIC) and High Income (HIC).

## 3. Results

Three major areas of cancer care that interfere with the economic aspects of the individual burden creating specific vulnerable groups can be differentiated: acute direct involvement with cancer diagnosis, cancer prevention programs (mainly investigated for cervical cancer) and indirect involvement (esp. families facing childhood cancer, cancer survivorship and family-based caregiving). The qualitative and quantitative research data are summarized in Table 1. In this context, vulnerability refers to financially related limits in the affordability, accessibility, availability and acceptability of cancer care.

Some investigations addressed, in part, economic consequences for the society, such as the loss of the workforce and the costs for outcomes (rehabilitation, long-term complications, survivorship issues, etc.). For example, the costs and cost-effectiveness of treating childhood cancers in LMICs [17] in terms of disability-adjusted life years (DALY), survival and country-specific life expectancy were recently compared with GDP products using the Consolidated Health Economic Evaluation Reporting Standards, although the identified body of evidence for these countries was still low and had high risks of bias, and true treatment costs were likely underestimated, Fung et al. [17] and Zabih et al. [18] concluded that overall childhood cancer treatment appears to be clinically effective and cost effective in LMICs, making sufficient cancer care achievable for these patients.

### 3.1. Affordibility

The affordability of cancer care has to be considered from two different perspectives: (A) the individual financial burden of the patients and their families due to cancer diagnosis and (B) the GDP-related affordability of cancer care for the society, which is mainly translated into limited availability of the respective resources. Financial consequences could arise due to the costs of treatment and the consequences of cancer.

Financial burden is a phenomenon that has been reported in many countries, contributing to the affordability of cancer care. For example, in the US, it was identified in about 50% of patients with cancer [28], and about half of this could be attributed to direct treatment costs [33]. In Germany, job incomes dropped more than one forth within one year after cancer diagnosis, whereas in China, belonging to UMICs, this thread can reach an entire year’s worth of GDP per capita [34]. The highest incidence and relative extent of catastrophic healthcare expenditures appears to involve cancer patients and their families when they already belong to low-income groups [35,36,37].

Table 2 provides an overview about the extent of financial difficulties or catastrophic health expenditures due to cancer diagnosis in directly involved cancer patients and in families affected by childhood cancer. Wide ranges result from investigations of different vulnerable groups and subpopulations, but in all available reports, the importance of financial burdens for cancer patients seems to be enormous. Financial catastrophe due to cancer was seen in many countries, which is in line with other non-communicable diseases (6–84% of the households, depending on the chosen catastrophe threshold) [15].

Overall, the SES appears to seriously determine the affordability and related consequences for cancer patients. The respective predictors of worse financial burdens were the lack of health insurance, lower income, unemployment and younger age at cancer diagnosis [26,50]. For example, vulnerable groups in industrial countries, such as African American [27] and Latino [51] cancer survivors in the US, disproportionately experience financial burden due to their disease. Especially in LMICs, as shown in India, the financial instability of cancer patients is of high importance [14].

However, the published evidence targeting the economic perspective of LMICs is biased towards costs incurred by the healthcare sector, and direct nonmedical as well as indirect costs were often not included [52]. Therefore, transferring results between countries is critical, but a loss of available income due to cancer seems to be a worldwide thread and important disease-related financial risks for these patients [14].

### 3.2. Accessibility

Besides the availability of healthcare resources, the socioeconomic determinants of healthcare usage can vary between regions, such as between African and Latin American LIMCs, due to the limited accessibility of required specialties [53]. Older and rural populations appear to be specifically endangered by this impaired access to cancer care [25,54]. The travel costs for treatment likely impact accessibility and affordability in vulnerable populations, such as elderly groups [12,55]. In addition, financial burden does not arise entirely from access to clinical care, but it also relates to access constraints due to indirect financial consequences and restricted social lives, causing additional travel costs, overnight accommodation or family reunions [22,23].

### 3.3. Financial Burden Affects Quality of Care

All stages of cancers, from prevention and early detections up to advanced cancer stages, appear to be at risk for financial burdens [42]. These burdens usually have an effect early in cancer treatment, independent from cancer sites, and they were associated with worse health-related QoL, nonadherence to cancer medication, shorter survival, poorer prognosis and a greater risk of recurrence [39,56]. The treatment abandonment of these patients was also impacted by the affordability of effective drugs and the availability of essential monitoring for its timely recognition. Older and rural adults are again particularly vulnerable and more likely to experience financial hazards, worsening their economic lifestyle during cancer survivorship [25,57]. For example, such an impact can result in twofold rates of disabilities and activity limitations [29]. Related risk factors include functional impairment, comorbidities, social support, impaired cognitive function and psychological state and financial stress [58].

Similarly, higher rates of treatment-related late effects and second primary malignancies [24], as well as reduced mental health and additional distress [31], are more likely to occur in financially challenged childhood cancer survivors. For example, nutritional factors with the interplay of malnutrition, the interference of one’s diet with drug absorption and the blood levels of cancer drugs seem to depend on financial conditions around the young cancer patients. [59] Moreover, the compliance of these patients with cancer appears to be influenced by environmental factors, such as the exposure to viral infections and pesticides, which may also be related to the socioeconomic framework.

The financial framework and the public and clinical facets of global cancer care appear to intensively interfere with the implementation of prevention and the early detection of cancer [60] and thus with the potential prognosis and outcome within the vulnerable groups. This area of cancer care is especially related to direct as well as indirect financial burden at the individual and societal level. Cervical cancer is used as an example for this tremendous impact [61,62], but the general picture can likely be transferred to other areas.

Despite worldwide efforts for HPV primary screening [63], the implementation rates are still not sufficient, and socioeconomic factors, available resources and acceptance issues contribute to this limitation [64]. Furthermore, vaccine availability enforces industrial countries into a moral dilemma due to the recommendations of extended target groups (boys) on one side and the overall shortage for the entire population in LMICs on the other [65]. In LMICs, screening programs have struggled with quality issues, and this is, at least in part, related to the economic aspects of healthcare implementation [66]. Their potential target population also frequently suffers from inadequate coverage [67] affected by reduced acceptance due the participants’ knowledge, attitudes and beliefs towards cervical cancer, which are to a large extent linked to individual economic factors and financial burden. For example, a higher age, a low educational status, a refugee/migrant or ethnic minority background, a menopausal status, housing conditions and a lack of insurance coverage appear to be linked with insufficient knowledge on the risk factors for cervical cancer. This results in false attitudes and perceptions on these preventive activities [21]. However, the empirical data investigating the relationship between individual socioeconomic aspects, the implementation of cancer prevention and subsequent negative effects on the quality of care and outcome, including the identification of special vulnerabilities due to the financial framework within the target groups, are not available yet.

Many countries have limited healthcare resources that can be made available for nationwide prevention programs, contributing to economic, political and societal instabilities. As a result of affordable national screening programs, the individual financial burden due to cancer will not only positively affect the direct costs of the prevention but will also improve indirect economic burdens, as described above [68].

### 3.4. Acceptability and Social Environment

The individual SES of cancer patients is intensively related to their acceptance of care, which is not limited to immediate clinical aspects. For example, perceived financial vulnerability appears as a determinant for insufficient perceptions of environmental and psychosocial cancer risk factors embedded in social and cultural contexts [69]. Disparities in cancer development and access to care are related to a large extent to those acceptance risks and protective factors that can be directly or indirectly attributed to the economic burden of the respective population, such as on the basis of racial and ethnic minority status, economic disadvantages, disability status, gender, geographic environment and nation of origin [41,48,70,71,72]. For childhood cancer survivors, important predictors of this vulnerability were female status, poor financial conditions, unemployment and poor education [73,74]. Furthermore, additional interference between economic burden and the acceptance of cancer care may be determined by varying socioeconomic milieu, such as families’ low SES, the long travel time, impaired family dynamics, the cancer center capacity, public awareness and governmental healthcare financing [46].

Besides the direct costs for cancer patients, their social environment also faces financial burdens due to the care for the involved person. Since, in most healthcare systems, this thread is not covered or even systematically addressed by insurance, etc., the extent of resulting consequences for cancer treatment and outcome have rarely been investigated so far. Social and economic deficits due to family-based caregiving for patients with cancer may include lifestyle disruption, less socializing, greater out-of-pocket payment and lost productivity costs [75]. Female partners are more vulnerable for these consequences, including personal life strain, social relations, financial burden and intrinsic rewards [23].

Pediatric cancer-induced financial distress and its adverse effects on parents is well documented [76]. The economic burden of these family members is not only an affordability barrier, but it also leads to reduced QoL and sickness of the family members, with potential worsening of the economic situation [32]. Moral distress within the families and reduced cancer care acceptance can result as consequence of the financial thread. The responsibility of healthcare providers to secure cancer care access for structurally vulnerable patients frequently relates to patients’ financial constraints and the resulting acceptability barriers to avoiding these conflicts [77].

The financial distress can likely induce cost-coping strategies at the individual level that interfere with treatment acceptance and compliance [78]. In addition, in industrial countries, supportive strategies, such as cancer care navigators or outreach programs, have been developed for the improvement of cancer care acceptance in these vulnerable patient groups [79]. Furthermore, social workers involved in the psychosocial treatment of cancer patients are requested to assess and address the financial and logistic aspects of life for comprehensive cancer care [26]. However, for LMICs, such strategies have not been reported and have likely not been implemented yet.

### 3.5. Availability

Cancer care usage in LMICs and, at least in part, in vulnerable groups in HIC is unevenly distributed throughout the different clinical specialties. For example, barriers to access, including inequalities in financial protection (mainly out-of-pocket payment), remain a fundamental challenge to providing surgical care or histopathological diagnostics [80]. Innovations in leapfrog technology and low-cost point-of-care tests may contribute to a reduction of the financial burden in LMICs [81].

For countries with limited economic resources for cancer care, various strategies have been recommended to provide the best care under the given conditions considering the existing financial framework (mainly targeting affordability and availability). Low-tech treatment protocols, such as switching from complex surgery to radiotherapy [82] or the usage of low-cost diagnostic procedures [83,84], were discussed. Even this availability in LMICs is very much limited [63,85,86] and correlates with the regional economic situation and outcome [53]. Additional examples were published for cervical [87] and pediatric cancer [88,89]. Alternatives might be the twinning of LMICs with high income nations [90,91,92], surgical mentorship, companion training programs [63] or the implementation of telemedicine [93,94] for the availability of specialty support.

These differences in diagnostic and treatment options, as determined by the given economic framework, are not limited to clinical availability but are furthermore related to participation in clinical research and access to innovation, respectively, due to financial limitations in these countries [95,96]. For example, in evaluations of breast cancer care, LMICs (2%) and Sub-Saharan Africa (9%) were grossly underrepresented [52]. However, access or acceptance bias in clinical research is also an issue in industrial countries, and vulnerable groups are likely disproportionately represented in cancer trials, resulting in severe selection as well as methodological bias. The restricted applicability of the evidence in LMICs provided by those trials that were only performed in certain areas worldwide and do not represent vulnerable groups accordingly leads to a worsening of cancer care, especially in these patient groups.

## 4. Discussion

More than a decade ago, the American Society of Clinical Oncology (ASCO) published their Guidance Statement on the Cost of Cancer Care, reflecting the increasing costs and financial burdens of cancer care, with a special focus towards the most vulnerable group, the cancer patients [97,98]. The addressed needs may currently have an even higher importance—not only for American patients but also for patients in other countries that face the incomplete coverage of cancer care by insurance or other backgrounds:Recognition cost of care discussions between the patient and physician as an important component of high-quality care;The provision of educational and support tools for cancer care providers promoting effective cost-related communication;Resource development to include the cost awareness of cancer care as part of shared decision making.

However, these requests are based on an existing patient–physician relationship and require that cancer patients have already found their way to cancer care. Therefore, the needs should be supplemented by bullet points related to public health challenges, such as for prevention and early detection:The identification and active targeting of vulnerable groups for cancer that are still outside the existing cancer care structures but should be addressed for improved prognosis and outcome.

Furthermore, numerous factors related to indirect costs for cancer care and determinants of financial vulnerability need to be considered:The recognition of different levels and reasons for financial or socioeconomical vulnerability as part of individual medical history.

The various aspects of the SES of cancer patients appear to be the most important determinants of their individual vulnerability regarding the affordability, accessibility and acceptability of cancer care. The same factors intensively affect the financial burden of these patient groups, resulting in the worsening of their cancer care coverage. In addition, disease-related reasons and public aspects can influence the financial burden and vulnerability of cancer patients (Figure 2).

Targeting vulnerable groups and their financial burden requires structured and comparable reporting of the entire cancer care processes. Usually, financial burden, economic hazards and related threads are solely investigated and discussed from the individual perspective of the cancer patients and sometimes of their social environment. However, there are additional economic effects on the society which are mainly out of focus when investigating cancer care. Reduced workforce availability, lower employment rates and premature retirement, among others, which are related to the direct cancer patients and their caregivers, may have negative consequences for a nation’s economy.

## 5. Conclusions

Non-communicable diseases, especially cancer, impose a substantial and growing global impact on families and impoverishment in all continents and at all income levels. The true extent, however, remains difficult to analyze due to the heterogeneity across existing studies in terms of the populations studied, the determinants considered, the outcomes reported and the measures employed. The impact that is exerted on the patients themselves, their families and their perspectives is likely to be underestimated. Important (socio)economic domains, such as indirect financial burden, economic handling and relief strategies and the inclusion of marginalized and vulnerable people who do not seek healthcare due to financial reasons, are underrepresented in the literature. Given the scarcity of information on specific regions, further research is required to estimate the impact of cancer diagnoses on households/families and impoverishment in LMICs, especially the Middle Eastern, African and Latin American regions [15]. However, the evaluation of financial burden, its determinants and its relationship with other aspects of the UHC criteria is not a phenomenon limited to these countries and has comparable importance for certain populations and patient groups in industrial countries.

The vulnerability of cancer patients due to financial burden and economic impact can be determined by various factors differing in certain subgroups, regions or countries. Similarly, the resulting consequences, such as treatment compliance, abandonment, the acceptance of care, impaired QoL, etc., depend on the specific environment of each cancer patient. This requires setting the right incentives to motivate all participating groups, including patients, healthcare providers, healthcare politicians and the society [99,100], and should be accompanied by evaluation strategies.

Research on cancer prevention, detection and management with a focus on vulnerable groups should not be limited to their specific risk factors, carcinogens and interactions between genetic and environmental factors, among other clinical and epidemiological topics [101]; rather, it requires additional efforts regarding implementation and outcome research to understand barriers and economic constraints for improved cancer care. This might require the adaptation of existing research methodologies, such as the consideration of regional distribution in trial recruitment, the potential worldwide transferability of evidence, modified or novel trial endpoints and statistical strategies for handling socioeconomic cofactors [102,103].

## Figures and Tables

**Figure 1 cancers-14-03158-f001:**
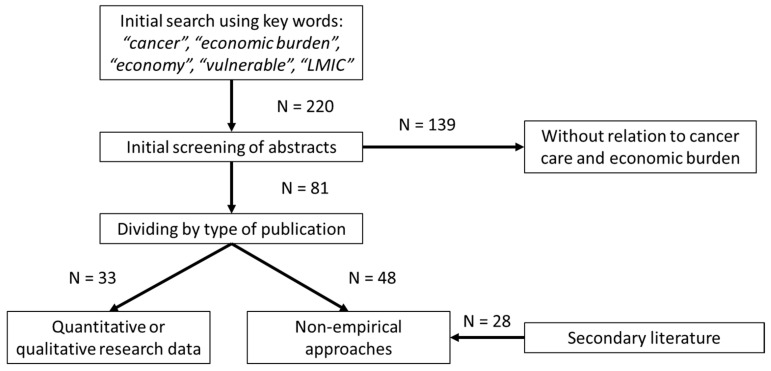
Literature search and evaluation strategy (CONSORT diagram).

**Figure 2 cancers-14-03158-f002:**
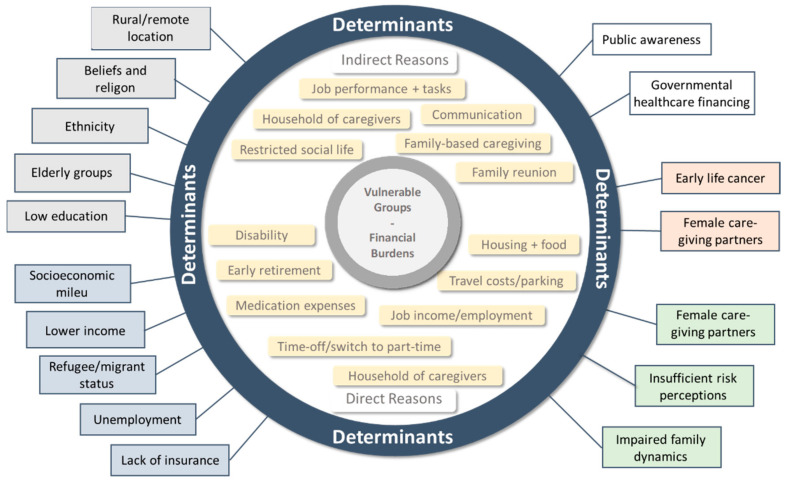
Direct and indirect reasons and determinants (grey: demographic; blue: economic; white: public; red: disease-related; green: social) of financial burden in vulnerable groups for cancer care.

**Table 1 cancers-14-03158-t001:** Quantitative and qualitative research approaches of the economic burden of cancer care (sorted by country and income status). Investigations that quantified individual financial consequences are highlighted in grey.

Reference	Country	Data Ressource	Method(s)	Key Results
**Western Industrial Countries**				
Parker et al. [12]	Australia (HIC)		Qualitative study, *N* = 11 acute myeloid leukemia (AML) in remission	Financial burdens: burden of AML-attributable costs (e.g., out-of-pocket parking, medication expenses); impact on paid work (e.g., early retirement, modifying job tasks); financial strain from AML (e.g., using savings, accessing government welfare), concerns about future familial financial burden (e.g., securing finances, worry about depleting financial resources).
Peretti-Watel et al. [19]	France (HIC)	Baromètre Cancer	National representative telephone survey, *N* = 3359	Those with a higher socioeconomic status (SES) are more likely to emphasize behavioral and psychosocial factors; those with an intermediate SES are more likely to affect environmental ones. Perceived financial vulnerability associated with higher perceptions for environmental and psychosocial factors.
Hernandez et al. [20]	Germany (HIC)	German Socio-Economic Panel survey	Comprehensive household surveys, *N* = ~20,000 individuals	Job incomes dropped 26–28% within one year after cancer diagnosis. Effect persisted for two years and was no longer observable after four years. Linked to increased likelihood of unemployment and reduction of working hours by 24%. Pension levels were not affected.
Riza et al. [21]	Greece (HIC)	Polyclinics run by two NGOs, participation in cervical cancer screening programs, free of charge	Cross-sectional study design, interviewer- administered questionnaire, *N* = 264 women	Behavior of women in relation with their knowledge, attitudes and beliefs towards cervical cancer and the HPV vaccine is categorized by predisposing factors (age, educational status, nationality, menopausal status, housing) and enabling factors (lack of insurance coverage). Older age, low educational background, refugee/migrant or ethnic minority (Roma) background, menopausal status, housing conditions and lack of insurance coverage linked with insufficient knowledge on risk factors for cervical cancer and false attitudes and perceptions regarding cervical cancer preventive activities (Pap smear and HPV vaccine).
Balfe et al. [22]	Ireland (HIC)		Qualitative analysis based on semi-structured interviews, *N* = 31 non-professional caregivers for patients with head and neck cancer	Financial impact on the household of caregivers during primary treatment in terms of travel costs, overnight accommodation, family reunion. Reduced household income due to changes in employment (reduction of working hours, giving up paid work). Long-term financial impacts are highly distressing.
Baanders et al. [23]	Netherlands (HIC)	Dutch Panel of Patients With Chronic Diseases	National population survey, *N* = 1093	Impact on social relations and financial situation in 20% of partners. Female partners more vulnerable for these consequences. Distinguished areas: personal life strain, social relations, financial burden and intrinsic rewards
Abrahão et al. [24]	United States (HIC)	California Cancer Registry	Cancer registry evaluation, *N* = 1168 adolescent and young adult cancer survivors after acute myeloid leukemia	Hispanic, Black or Asian/Pacific Islander (vs. non-Hispanic White) races/ethnicities and those who resided in lower SES neighbourhoods were at a higher risk of numerous late effects, including endocrine (26.1%), cardiovascular (18.6%), respiratory (6.6%), neurologic (4.9%), liver/pancreatic (4.3%), renal (3.1%), avascular necrosis (2.7%) and second primary malignancies (2.4%).
Azuero et al. [25]	United States (HIC)	Rural Breast Cancer Survivors Study	Population-based survey, *N* = 331 breast cancer survivors	Physical health status was predicted by BMI, comorbid conditions, social support and adverse changes in economic lifestyle in older rural breast cancer survivors (55–90 y).
Callahan et al. [26]	United States (HIC)	Financial Social Work Initiative	Cross-sectional survey, *N* = 90 cancer patients	Health insurance adequacy, fewer perceived barriers to care and reduced financial stress were significant predictors of better financial quality of life.
Hastert et al. [27]	United States (HIC)	Detroit Research on Cancer Survivors Cohort	Population-based cross-sectional survey, *N* = 916 African American breast, colorectal, lung and prostate cancer survivors	Nearly half of employed survivors changed employment framework due to cancer; 34.6% took at least one month off of work, including 18% with unpaid time off. More survivors employed full time (vs. part time) at diagnosis were on disability (18.7% vs. 12.6%, *p* < 0.001), while fewer were unemployed (5.9% vs. 15.7%, *p* < 0.001). A total of 47.5% of employed survivors decreased work participation. Unpaid time off, but not paid time off, was associated with decreased work participation.
Ko et al. [28]	United States (HIC)	Urban region, 3 outpatient cancer facilties, inner-city hospital	Cross-sectional survey, *N* = 104 patients in ambulatory cancer care services	A total of 77% reported concerns with one or more socio-legal need in the past month (mean: 5.75 concerns per participant). Most common socio-legal concerns related to income supports, housing and employment/education.
Lu et al. [29]	United States (HIC)	Ongoing, national, cross- regional, long-term, annual family interview survey of civilian non- institutionalized population	Cross-sectional study using data from the National Health Interview Survey, *N* = 15,002,192 cancer survivors	Cost-related medication nonadherence was associated with increased economic burden (OR: 1.89, 95% CI: 1.70–2.11), increased bed disability day (IRR: 1.46, 95% CI: 1.21–1.76), activity limitation (OR: 1.42, 95% CI: 1.25–1.60) and functional limitation (OR: 2.12, 95% CI: 1.81–2.49).
Nedjat-Haiem et al. [30]	United States (HIC)		Cross-sectional study, Functional Assessment of Cancer Therapy-General (FACT-G) questionnaire, *N* = 68 latinos	For older Latinos with chronic diseases (incl. cancer), financial hardship was associated with worse QoL. Financial hardship and financial worry were the most important covariates for treatment adherence.
Perez et al. [31]	United States (HIC)	Childhood Cancer Survivor Study	Multi-institutional, retrospective cohort study, *N* = 698 childhood cancer survivors	History of distress predicts lack of mental health support (*p* = 0.60) and is also related to uninsured status and cost coverage restrictions (*p* < 0.001). Males (OR = 2.96) and survivors with public (OR = 6.61) or employer-sponsored insurance (OR = 14.37) were more likely to have mental health coverage. Most vulnerable survivors, specifically those uninsured and with a history of distress, are at a risk of experiencing challenges accessing mental healthcare.
Santacroce et al. [32]	United States (HIC)		Investigator-developed online survey, *N* = 87 fathers of children with cancer	Pediatric cancer-induced financial burden contributed to fathers’ symptom severity and burden and QoL declines.
Tangka et al. [33]	United States (HIC)	California, Florida, Georgia and North Carolina population-based cancer registries	Cross-sectional survey, *N* = 830 women under 40 years of age diagnosed with breast cancer	A total of 92.5% of respondents were continuously insured (past 12 months); 9.5% paid a “higher price than expected” for coverage. Common concerns among 73.4% of respondents employed at diagnosis included increased paid (55.1%) or unpaid (47.3%) time off, suffering job performance (23.2%) and staying at (30.2%) or avoiding changing (23.5%) jobs for health insurance purposes. A total of 47.0% experienced financial decline due to treatment-related costs. Patients with some college education, multiple comorbidities, late-stage diagnoses and self-funded insurance were most vulnerable.
**Asian Countries**				
Li et al. [34]	China (UMIC)	Hua County, Henan Province	Inpatient claim data (Rural Cooperative Medical Scheme), *N* = 2375 for esophageal, gastric and colorectal cancer	For each hospitalization to treat esophageal cancer, the average total cost and out-of-pocket expenses after reimbursement equaled the entire year’s local GDP per capita and disposable income per capita. Usage depends on age (decreasing over 60 y) and gender (more females in younger ages)
Sui et al. [35]	China (UMIC)	Two tertiary hospitals	Cross-sectional interview, *N* = 242 households living with pediatric leukemia	Overall incidence of catastrophic health expenditure was 43.4% (lowest income group 69.0%, highest 16.1%). Medical insurance, frequency of hospital admissions, charity assistance and income level were significant predictors.
Sun et al. [36]	China (UMIC)		Hospital-based multicenter retrospective survey *N* = 470 households with lung cancer patients	Health insurance protects some households from the impact of catastrophic health expenditures. Its incidence (78.1%) and intensity (14.02% for average distance and 22.56% for relative distance) are relatively high among households with lung cancer patients. Incidence is lower in households covered by the Urban Employee Basic Medical Insurance (UEMBI), with higher income levels and shorter disease courses.
Sun et al. [37]	China (UMIC)		Multicenter, cross-sectional interview surveys, *N* = 639 households with breast cancer patients	Mean out-of-pocket expenditure accounted for ~55.20% of mean households’ non-food expenditures. Overall incidence of catastrophic health expenditures was 87.95 and 66.28% before and after insurance compensation, respectively. Education, disease course, health insurance, treatment method and income were significant predictors.
Kastor et al. [38]	India (LMIC)	National Sample Survey Organization	Survey on disease-specific financial distress, *N* = 333,104 individuals from 65,932 households (36,480 rural, 29,452 urban)	About 28% of households incurred catastrophic health expenditures and faced distress financing. Among all diseases, cancer caused the highest catastrophic health expenditure (79%) and distress health financing (43%). Likelihood of incurring distress financing higher for those hospitalized for cancer (OR 3.23; 95% CI: 2.62–3.99).
Wajid et al. [14]	India (LMIC)	Bengaluru	Qualitative interviews, *N* = 10 advanced cancer patients	Prevalent problems were financial instability, hopelessness, family anguish, self-blame, helplessness, anger, stress and suicidal thoughts.
Lim et al. [19]	Malaysia (UMIC)		Routine clinical surveillance for hypothetical cohort, *N* = 1000 BRCA testing in early-stage breast cancer patients	Testing generated 11.2 QALYs over the lifetime and cost USD 4815 per patient, whereas routine clinical surveillance generated 11.1 QALYs and cost USD 4574 per patient. Incremental cost-effectiveness ratio was below cost-effective thresholds.
Malhotra et al. [39]	Singapore (HIC)	COMPASS cohort study	Cost of medical care data, *N* = 600 stage IV solid malignancy patients	A total of 35% had difficulty in meeting expenses. A higher financial difficulties score was associated with worse physical, psychological, social and spiritual outcomes and a lower perceived quality of healthcare coordination and responsiveness (all *p* < 0.05), persisting after adjustment for SES indicators.
**African and American Countries**				
Bhakta et al. [40]	Brazil (UMIC), Malawi (LIC)		Disability-adjusted life years (DALYs) cost-effectiveness thresholds WHO-CHOICE, Brazil *N* = 1344, Malawi *N* = 447 national incidence for ALL and Burkitt lymphoma	3:1 cost/DALY to GDP/capita ratio for ALL in Brazil was USD 771,225; expenditures below cost effective threshold. Costs below USD 257,075 (1:1 ratio) very cost effective. BL in Malawi USD 42,729 and USD 14,243, resp. Actual costs Brazil, USD 16,700; Malawi total drug costs less than USD 50 per child.
Rativa Velandia et al. [41]	Colombia (UMIC)	Vulnerable population in Bogotá	Descriptive, cross-sectional study, *N* = 50 families of children with cancer	Childhood cancer families had a high economic burden: transportation (28.5%), communications (26.3%), health (20.8%), housing (19.7%), food (17.4%).
Unger-Saldaña et al. [42]	Mexico (UMIC)	Four major public cancer hospitals in Mexico City	Cross-sectional survey, *N* = 886 breast cancer patients	Diagnostic interval was longer among women who were single, interpreted symptoms as not worrisome, concealed symptoms and perceived lack of financial resources and difficulty of missing work as barriers to seeking care. Barriers more commonly perceived among patients who were younger, lower socioeconomic status and living outside Mexico City
Agulnik et al. [43]	Guatemala (UMIC)	National Pediatric Oncology Unit	Hospital administrative data, implementation costs, *N* = 4334 hospital admisisons pediatric cancer patients	Variable costs of unplanned pediatric intensive care unit transfer versus regular ward was GTQ 806 per day. Total cost of implementing pediatric early warning systems was GTQ 7 per admission.
Denburg et al. [44]	Uganda (LIC)	Uganda Cancer Institute	DALYs, cost-effectiveness thresholds WHO-CHOICE, *N* = 122 pediatric Burkitt lymphoma patients	The cost per DALYs averted in the treatment group was USD 97; national DALYs averted through treatment was 8607 years. Cost were within WHO-CHOICE cost-effectiveness thresholds. The ratio of cost per DALY to per capita gross domestic product was 0.14, reflecting a very cost-effective intervention.
**Multinational Investigations**				
ACTION Study Group. [45]	8 LMIC in Southeast Asia	ASEAN Costs in Oncology Study	Routine clinical surveillance for hypothetical cohort, *N* = 9513 newly diagnosed cancer patients	Economic hardship reported by a third of families, including an inability to pay drugs (45%), mortgages (18%), utilities (12%); 28% taking personal loans; 20% selling assets. Households initially above national poverty levels: 49% pushed into poverty at one year. In all countries, the cancer stage largely explained the risk of adverse outcomes.
Friedrich et al. [46]		101 countries	Treatment abandonment survey for pediatric cancer, *N* = 581 healthcare staff	LMIC considered SES factors (families’ low SES status, low education and long travel time) as most influential in increasing the risk of treatment abandonment. Emerging factors: vulnerability, family dynamics, perceptions, center capacity, public awareness and governmental healthcare financing, among others.
Manchanda et al. [47]	UK/USA/Netherlands (HIC), China/Brazil (UMIC), India (LMIC)		Hypothetical cohort, lifetime costs and effects of BRCA1/BRCA2 testing on all general population women ≥30 years	Incremental cost-effectiveness ratio (ICER)/QALY: societal perspective—cost-saving in HIC, cost-effective in UMIC, not cost-effective in LMIC; payer- perspective: highly cost-effective in HIC, cost-effective in UMIC, not cost-effective in LMIC. BRCA testing costs below USD 172/test (ICER = USD 19,685/QALY), which is cost-effective (from a societal perspective) for LMIC/India. Testing can prevent an additional 2319 to 2666 breast cancer cases and 327 to 449 ovarian cases per million women.
Raja et al. [48]		Cancer Incidence in Five Continents (CI5)	Cancer incidence data, Childhood brain tumors (CBT)	Incidence is the highest in North America and the lowest in Africa. CBT incidence rates increased significantly with increasing GDP per capita (*p* = 0.006). Gini index is significantly negatively associated with CBT incidence. Incidence decreased with increasing income inequality within countries (higher Gini indices, *p* = 0.040).
Tangka et al. [49]	Uganda (LIC), Kenya (LMIC), India (LMIC), Colombia (UMIC)	Uganda (Kampala), Kenya (Nairobi), India (Mumbai), Colombia (Pasto, Barranquilla, Cali, Bucaramanga, Manizales)	Cancer incidence data, eight population-based cancer registries, Center for Disease Control and Prevention’s International Registry Costing Tool (IntRegCosting Tool),	Cost per cancer case registered: LIC and LMIC (USD 3.77 to USD 15.62); UMIC (USD 41.28 to USD 113.39). Registries serving large populations (over 15 million inhabitants) had a lower cost per inhabitant (<USD 0.01 in Mumbai, India) than registries serving small populations (under 500,000 inhabitants) [USD 0.22] in Pasto, Colombia.

WHO-CHOICE: WHO Choosing Interventions That Are Cost-Effective; QoL: Quality of Life.

**Table 2 cancers-14-03158-t002:** Financial impact of cancer diagnosis on patients and their families in different countries.

Country	Percentage of Cancer Patients Facing Financial Difficulties or Catastrophic Health Expenditures *	
	Direct Cancer Patients	Childhood Cancer Families ^2^ Family-Based Caregiving ^3^
Germany (HIC)	26–28%	
Singapore (HIC)	35%	
United States (HIC)	34–77%	
China (UMIC)	16–88%	
India (LMIC)	79%	
ASEAN group (UMIC & LMIC)	12–45%	
Netherlands (HIC)		20% ^3^
Colombia (UMIC)		17–28% ^2^

* Definitions of financial difficulties or catastrophic health expenditures vary between the various investigations.

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
