# Peer review of "Economic Perspective of Cancer Care and Its Consequences for Vulnerable Groups"

_cancers, 2022, doi:10.3390/cancers14133158_

Round 1
Reviewer 1 Report
See attchment.

Author Response
Thank you very much for the review and comments.
We agree with the reviewer that we did not prepare a classical review. We tried to make this clear by choosing the "narrative review" format since other categories for manuscripts that are more suitable for the topic of this invited manuscript are currently not available for the journal. By targeting the proposed topic it was clear that typical approaches for a review, such as for clinical evidence, based on published data with detailed comparison, describing ranges etc. is not achievable due to the extreme heterogeneity of the published information and results. However, systematic search and critically summarizing the currently available evidence and information about economically vulnerable groups is more than a political statement.
We do not agree with the feedback that the critical summary of the published findings only provides common knowledge statements. Although vulnerability for special groups of patients is gaining more and more importance its characterization and the determinants of this vulnerability are just at the beginning of generating related evidence and facts. From our perspective it is very important to replace the individual perception that all these facts are of similar importance with evidence. Our aim was to review which aspects should have priorities for improvement of cancer care, esp. in these countries.
The economic background of the countries that were listed and discussed is now better included in table 1. Grouping was done into Western industrial countries, Asian countries, African and American countries and multinational investigations. Within the groups the countries were listed alphabetically and for multiple studies from one country in the alphabetical order of the first author. Within the table the column “Methods” was reorganized and missing information was added: Method of data acquisition; samples size; target population. Country and its grouping by income was added as new column.
The requested data regarding GDP spend for cancer care or overall expenses per patient are not available as it would be required to take this as a basis of evaluation. Even in high income countries, most of the costs have not been adressed yet and it is one of the aims of this review to summarize the aspects that need to be considered for an overall picture of economic burden. And various aspects of financial issues, such as its perceived burden, cannot be counted moneywise and limitation to reimbursement data would result in underestimation. Overall, focussing on GDP-related numbers has a high risk of providing an incomplete picture of the real burden.
References were cited in the manuscript as related to the context. Where appropriate they are listed directly referring to single information, but in various cases we summarized given information and in these cases the reference was provided at the end of a sentence.
We do not understand the proposed link between economic burden and financial vulnerability at the individual level with cancer excess death rates due to the pandemic. There are various arguments to avoid a mixture between these two topics.
- trusted, population-based evidence of these excess rates is still very limited and single institution report should not be used as indicators for resilience of entire healthcare systems
- it is highly questionable whether such numbers are suitable indicators for resilience in general, eg. due to missing methodological standards, data availablility in many countries etc., but also large numbers of cofactors in a pandemic
- from our perspective proposing cancer care management during pandemic as clue for a ranking ("defining the best") of healthcare systems appears to be highly speculative
Language comments were adressed and the manuscript was carefully checked and modified as appropriate.
Reviewer 2 Report
This paper attempts to provide a narrative review on an important, yet understudied topic. Unfortunately, it requires major revision to be publishable. The first issue relates to readability – the text is lengthy, uses undefined terms and lacks a cohesive, easy to follow structure. The methods are largely undefined, which makes it difficult to critically evaluate the robustness of the terms used and the statements made. Therefore, while I think this paper has great potential for this special edition of Cancers, it requires significant editing to ensure it contributes to the evidence base and can be used to guide future research and practice.

Author Response
Thank you very much for the review and the suggestions.
According to the conceptual concerns we rearranged the entire result and discussion sections. As suggested the flow of the presentation is now organized along the UHC criteria. The examples for which more publications were available (childhood cancer and cervical cancer screening) were brought into a better context within this concept and we did not present them as separate subsections anymore. Furthermore, they were shortened.
The search and evaluation strategy for the literature was included in a new figure 1. Additional methodological information regarding the grouping of the countries was provided and considered as new column in Table 2 (former table 1).
Furthermore, lengthy parts of the manuscript were shortend.
Reviewer 3 Report
This review examines economic perspectives of cancer care and their consequences for vulnerable groups. The focus of the review is cancer care in Low-Middle-Income-Countries and in other vulnerable groups due to direct or indirect cancer care costs.
The topic is extremely important for cancer care - with healthcare costs rising, demographic and lifestyle changes worldwide, and the healthcare disparities gap widening the need to understand vulnerable groups in cancer care due to economic consequences is paramount.
Despite the enthusiasm for the importance of the topic, the review suffers from methodological and conceptual shortcomings.
Methodological concerns:
The review needs to be reproducible: please clarify the exact research question, provide the search terms, keywords, inclusion and exclusion criteria of primary studies, methodological details for data extraction, and a consort diagram for the identified studies and reasons for exclusion.
Table 1. needs significant re-organization so that the reader can easily identify the primary studies and navigate the table to find relevant information. Studies can be listed alphabetically based on authors' last name. Alternatively, countries may be listed alphabetically, and within each country studies can be listed based on authors' last name. The information presented under the column termed "Methods" is extremely heterogeneous. For some studies the reader finds only the sample size of the primary study, for other studies design is included. List consistently the same information for each primary study: question, design, sample etc.
Conceptual concerns:
The logic with which authors organize studies, synthesize evidence, and present findings is not clear and the reader has a hard time following the flow of the manuscript. Currently, findings present costs of direct cancer care and family caregiving with focus on childhood cancers. Then the authors present cervical cancer as an example, where costs are associated primarily with primary prevention and vaccination and concern the society and the healthcare system. This is very different from catastrophic healthcare expenditures related to childhood cancers mentioned above.
This reviewer suggests that a possible organization of findings can be based on universal healthcare coverage criteria: availability of services, acceptability, accessibility, affordability and quality of care, that the authors mention in the Introduction, and identifying vulnerable populations within each criterion. Alternatively, the authors may focus on specific types of cancer, possibly guided by worldwide incidence and prevalence, and organize the review accordingly.
Along these lines, Figure 1 needs to present information in a more organized way, for example, individual factors (demographics, psychosocial etc.), healthcare system factors (prevention, surveillance, care delivery, etc.)
The Discussion presents information that should be mentioned under Results for example, the discussion about lack of histophathological services
Issues of lesser importance that need attention:
Introduction, lines 46 - 51, please add family caregiving to the list of indirect care costs
Lines 136-137, please edit for clarity. The same for lines 140-143.
In line 145 the authors use the term "coping strategies" to refer to responses from the healthcare system, which is not the correct use of the term. Coping strategies refer to the individual (person) psychological responses aiming to mitigate an internal or external stressor. Please edit this term, which it is used more than once in the manuscript.
Author Response
Thank you very much for the review and the suggestions.
According to the conceptual concerns we rearranged the entire result and discussion section. As suggested the flow of the presentation is now organized along the UHC criteria. The examples for which more publications were available (childhood cancer and cervical cancer screening) were brought into a better context within this concept and we did not present them as separate subsections anymore. Furthermore, they were shortened.
Table 1 was reorganized and additional structure was included. Grouping was done into Western industrial countries, Asian countries, African and American countries and multinational investigations. Their economic status is now provided according to the Woldbank ranking. Within the groups the countries were listed alphabetically and for multiple studies from one country in the alphabetical order of the first author.
For all references the column “Methods” was reorganized and missing information was added: Method of data acquisition; samples size; target population. Country and its grouping by income was added as new column.
Figure 2 (former figure 1) has been adapted according to the suggestions which can be found in the legend: Direct and indirect reasons and determinants (grey: demographic; blue: economic; white: public; red: disease-related; green: social) of financial burden in vulnerable groups for cancer care
“Family-caregiving” has been added to the list of indirect costs.
The questioned paragraph (lines 136-143) has been modified accordingly for better clarity and included in the rearrangement of the manuscript.
The term coping has been brought in the correct context or replaced.
Round 2
Reviewer 3 Report
The authors have been responsive in addressing Reviewers' comments. However, there is still one issue pending and that relates to Figure 2. In the present form, it is unclear what is the contribution of Figure 2 to the manuscript. The organization of findings in the Results section do not appear in Figure 2 and the reader struggles to understand its contribution to the overall conceptualization of the manuscript. Is there any other way the authors can organize Figure 2?
Also in the Discussion and Conclusions it is important for the authors to discuss how many of these findings apply to countries with national health insurance programs. Bringing this argument into light will increase the significance of the manuscript. If this information is already included, the reviewer missed it.
Author Response
Thank you very much for the comments. We want to reply as follows:
Figure 2 has been moved to the discussion to bring it into a better context of the content. We included a paragraph that better provides the link between the determinants of financial burden and vulnerability for cancer patients.
We fully agree that the potential link between national health insurance programs and the financial vulnerability of cancer patients is an important topic. However, it was difficult to adress this in the manuscript for several reasons:
- quantitative investigations are currently only available for very few countries
- availability of health insurance systems does not necessarily mean that the patients have access to this. Furthermore, the structure differs extensively and is very difficult to compare. Therefore, we did not address this topic in the manuscript